# The use of a speaking book® to enhance vaccine knowledge among caregivers in The Gambia: A study using qualitative and quantitative methods

Oluwatosin O Nkereuwem [1], Sonali Kochhar,[2,3] Oghenebrume Wariri [1], Penda Johm,[1] Amie Ceesay,[1] Mamanding Kinteh,[4] Beate Kampmann [1,5]

For numbered affiliations see end of article.

**Correspondence to**
Professor Beate Kampmann;
bkampmann@mrc.gm

## ABSTRACT

**Objectives** To measure the usefulness of a Speaking Book (SB) as an educational tool for enhancing knowledge, understanding and recall of key vaccine-related information among caregivers in The Gambia, as well as its acceptability and relevance as a health promotion tool for caregivers and healthcare workers.

**Design and setting** We developed a multimedia educational tool, the vaccine Speaking Book, which contained prerecorded information about vaccines provided in The Gambia's Expanded Programme on Immunization. Using qualitative and quantitative methods, we then conducted a sequential study assessing the use of this tool among caregivers and healthcare workers in The Gambia.

Participants
200 caregivers attending primary healthcare centres in The Gambia for routine immunisation services for their infants, and 15 healthcare workers employed to provide immunisation services at these clinics.

**Outcome measures** We calculated the median knowledge scores on vaccine-related information obtained at baseline, 1-month and 3-month follow-up visits. Wilcoxon's matched-pairs signed-rank test was used to compare the difference in the median knowledge scores between baseline and 1-month, and between baseline and 3-month follow-up visits.

**Results** Of the 113 caregivers who participated, 104 (92%) completed all three study visits, 108 (95.6%) completed the baseline and 1-month follow-up visits, and 107 (94.7%) completed the baseline and 3-month follow-up visits. The median knowledge score increased from 6.0 (IQR 5.0–7.0) at baseline to 11.0 (IQR 8.0–14.0) at 1-month visit (p<0.001), and 15.0 (IQR 10.0–20.0) at 3-month visit (p<0.001). Qualitative results showed high acceptability and enthusiasm for the Speaking Book among both caregivers and healthcare workers. The Speaking Book was widely shared in the community and this facilitated communication with healthcare workers at the primary healthcare centres.

**Conclusions** Context-specific and subject-specific Speaking Books are a useful communication and educational tool to increase caregiver vaccine knowledge in low/middle-income countries.

## Strengths and limitations of this study

► The vaccine Speaking Book (SB), a richly illustrated, context-specific, audio-visual, educational tool was specifically designed to complement vaccine information given to caregivers by healthcare workers in immunisation clinics in The Gambia.
► The qualitative and quantitative approach employed in this study explored the impact of the SB as a health promotion tool, involving both caregivers and healthcare workers.
► A limitation of this study is a possibility of self-selection bias due to the absence of a control group.
► Although we encouraged the study participants to share the SB with their family and friends, we did not capture any effects on the overall vaccine knowledge at the community level.
► We did not systematically measure the impact on subsequent uptake of immunisation services.

## INTRODUCTION

Immunisation is one of the most effective public health interventions for the control and prevention of infectious diseases.[1] Unfortunately, global immunisation coverage rates remain suboptimal with approximately 21.8 million infants never completing the recommended immunisation schedule.[2 3]

Several factors may act as barriers to childhood immunisation especially in low/middle-income countries (LMICs) where caregivers often have low literacy levels.[4–7] In addition to the wider concerns of vaccine confidence that have been expressed across the globe, more specific issues include caregivers' lack of knowledge about the recommended immunisation schedules and alternatives once their infants miss scheduled immunisations, concerns about vaccine safety and adverse events, infants being unwell at the time of the appointment, reluctance to receive multiple vaccinations at the same time, lack of trust in the medical community, being unable to

remember the information provided and the date for the next immunisation visit, hesitancy in asking questions which might be perceived as trivial by healthcare workers (HCWs), and HCWs being too busy or lacking the knowledge or patience to answer the caregivers' questions and addressing their concerns.[4 8–10]

The caregivers' knowledge and overall vaccine knowledge influence their decision to access immunisation services for their infants.[8 11] This decision in turn impacts on the uptake of vaccination services, and consequently, on the morbidity and mortality attributable to vaccine-preventable diseases.[8 9] Data from high-income countries (HICs) have shown that improving health awareness and knowledge of caregivers about disease prevention can improve health outcomes, especially among less literate populations.[12] This would be applicable especially in LMICs in sub-Saharan Africa where caregivers have been shown to have low knowledge about immunisation and vaccine-related issues.[13]

The Gambia's Expanded Programme on Immunization (EPI) is considered as highly successful compared with other countries in sub-Saharan Africa. The programme has consistently maintained coverage of the third dose of diphtheria–tetanus–pertussis (DTP3) above 95% and DTP1 to DTP3 dropout rates below 10% since 2005. However, despite these overall high coverages documented over the years, there still exist wide intracountry equity gaps along the lines of caregivers' education and wealth quintile.[14]

Recently, audio-visual tools like the Speaking Book (SB) have been developed and used to target HCWs and care-seekers in HIV care and mental health services, with studies demonstrating their usefulness in the improvement of knowledge.[12 15 16] These studies have also shown that SB complements communication between HCWs and healthcare seekers. In a previous study conducted in The Gambia, a clinical trials-specific SB was shown to increase the knowledge of study participants, leading to a better informed decision about participation.[12] Therefore, using context-specific, pictorial depictions followed by recorded audio messages targeted at caregivers are considered an effective way to deliver messages about vaccines in general, especially in a low-literacy setting such as The Gambia.[17]

In this study, we developed the vaccine SB, a new SB covering routinely recommended maternal and childhood vaccines. We assessed its use as an educational tool for enhancing knowledge, understanding and recall of key vaccine-related information among caregivers in The Gambia, as well as its acceptability and relevance as a health promotion tool for caregivers and HCWs.

## METHODS
### Study design and setting
We conducted a study which used quantitative and qualitative methods to enrol caregivers and their infants attending immunisation clinics in 15 purposively selected primary healthcare facilities (PHCs) across four regions of The Gambia. The PHCs comprised of seven rural and eight urban centres that had not previously participated in vaccine-related research studies or clinical trials. Based on existing records, the Gambia EPI selected immunisation facilities with poorer performance compared with expected national outcomes to participate in the study. The project was conducted in close collaboration with the communications department of the Gambia EPI.

Immunisation services are provided free-of-charge at all government facilities in The Gambia, and caregivers are encouraged to access the services closest to their home. Immunisation-related information should be routinely delivered during immunisation clinic days by public health officers (PHOs). For the purpose of this study, we defined a caregiver as an adult aged 18 years and above attending the immunisation clinic with an infant, and responsible for that infant's day-to-day care. This included but was not limited to biological parents.

### Study tool
The prototype of the richly illustrated, audio-visual educational tool called a Speaking Book was developed by a US-based company (https://speakingbooks.com/), with whom we collaborated. The SB concept can be adapted for specific purposes, and in our case, we adapted the text, recording and illustrations to vaccines routinely delivered by the Gambia EPI programme.

During an iterative pilot phase, we developed a bespoke SB version and conducted four separate focus group discussions (FGDs) with the Gambia EPI programme managers, HCWs and caregivers attending EPI clinics to ensure that the final version of the SB reflected the local Gambian context. The final version of the SB was an A4-sized hard cover book consisting of 16 pages of colourful, culturally sensitive illustrations with short texts written in English and recorded narrations in the two most widely spoken local languages, Wolof and Mandinka. Each SB has a plastic panel with removable battery, which hosts a series of push buttons, each corresponding to a specific page in the SB. When activated, the push buttons trigger a soundtrack of the text on the relevant page. The soundtrack was narrated by two respected local actors, with the appropriate voice and tonal quality. The language could be selected via a switch button. Figure 1 shows some photographs of the SB (see online supplemental material 1).

### Data collection methods
#### Quantitative methods
Following a wide literature search, we designed a structured questionnaire for the quantitative data collection. To assess the construct and content of the data collection tool, we pretested it with 25 caregivers and 5 HCWs who provided feedback which was used to refine the wording of the questions and response options. The final version of the questionnaire consisted of five sections capturing the following: (1) sociodemographics of the caregiver (19

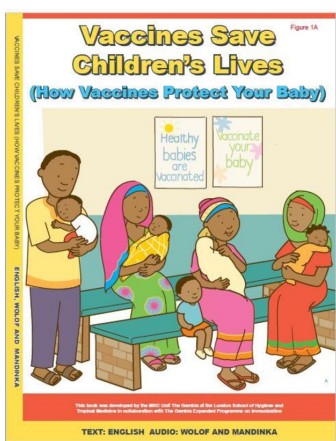
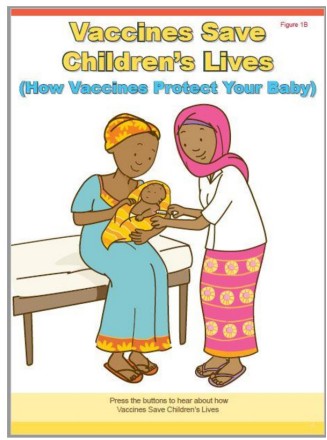
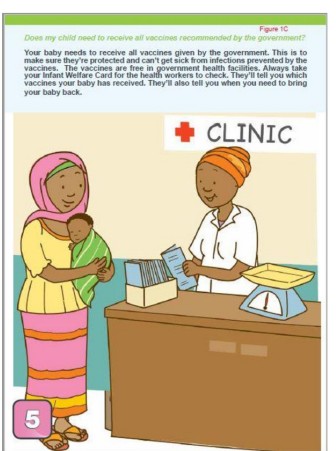
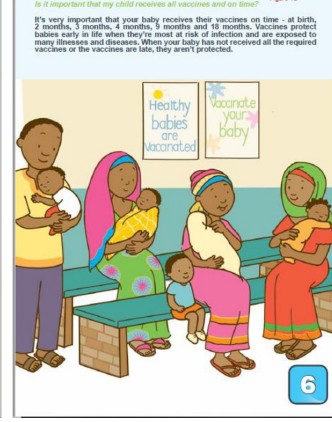

**Figure 1** Photo of the Speaking Book.

questions); (2) immunisation history and experiences of the caregiver (five questions); (3) sociodemographics of their child(ren) (10 questions); (4) a series of 8 multiple-choice questions with 39 correct choices assessing vaccines and immunisation knowledge of the caregiver; and (5) a series of 18 open and close-ended questions assessing the experiences of the caregiver on the use of the SB. A sample questionnaire is included in online supplemental materials 2 and 3.

### Qualitative methods

A qualitative approach was used to explore the perception of HCWs on the use of the SB as a health promotion tool when sharing vaccine information with caregivers during immunisation clinics. We carried out in-depth interviews (IDIs) with HCWs using 11 general, open-ended and non-leading questions. These included questions which explored the challenges they faced when sharing vaccine information with caregivers, impact of the SB tool on their work, outcomes since the utilisation of the SB and suggestions on the future use of the tool. The interview guide is in the online supplemental material 4.

### Participant selection and data collection

All study participants were sensitised on the study procedures and objectives. Participants were eligible for inclusion if they were caregivers with infants below 6 months of age attending one of the selected PHC centres in The Gambia, had been living at their current address for at least 6 months, were able to communicate in either Wolof or Mandinka, and were able to provide written consent. Purposive sampling strategy was used to select caregivers who met the inclusion criteria for enrolment into the study.

During an initial sensitisation visit, the research staff educated the caregivers on the aim of the study and gave the caregivers an information sheet and consent form to take home and return on another date. Each caregiver who returned a completed consent form was then invited to participate in a baseline visit during which the questionnaire was interviewer administered to obtain socio-demographic information and to assess their baseline knowledge about childhood and maternal vaccines. Only one attempt was allowed for response to the questions. To assess the knowledge of caregivers, we computed a knowledge score which was calculated by assigning a score of 1 for each correct answer and 0 for each incorrect answer, with maximum and minimum knowledge scores of 39 and 0, respectively. Only sections 1–4 of the questionnaire were administered during this baseline visit.

Subsequently, the mechanics of the SB were explained, and each caregiver was given a copy of the SB for use at home. Each PHC head was also given a copy of the SB to be used during health education activities in the facilities. Caregivers could listen to the SB as many times as they wished during the entire study period and were also encouraged to invite other people to listen along with them. Following receipt of the SB, the participants were requested to return to the PHCs for follow-up visits at 1 month and 3 months after the baseline visit. During each of these follow-up visits, the participants' understanding, retention, utilisation of key information and experiences using the SB were assessed with the same questionnaire used at baseline. Section 5 of the questionnaire was also administered during this visit.

At the end of the 3 months, we conducted IDIs with one representative HCW from each of the participating immunisation clinics. During this interview, we assessed their perception of the acceptability, potential efficacy and use of the SB as a health promotion tool for HCW delivering immunisation services in The Gambia. Qualitative data were audio-recorded and transcribed by the research team.

### Data analysis

The median knowledge scores were calculated at the baseline, 1-month and 3-month follow-up visits. We compared the difference in the median knowledge scores between baseline and 1 month, and between baseline and 3 months for the entire cohort and by subgroups using the Wilcoxon's matched-pairs signed-rank test for non-parametric paired data. The subgroups used for analysis included region, age, household income and level of education of caregiver. The distribution of the scores is presented in tables showing median scores and IQRs as appropriate,

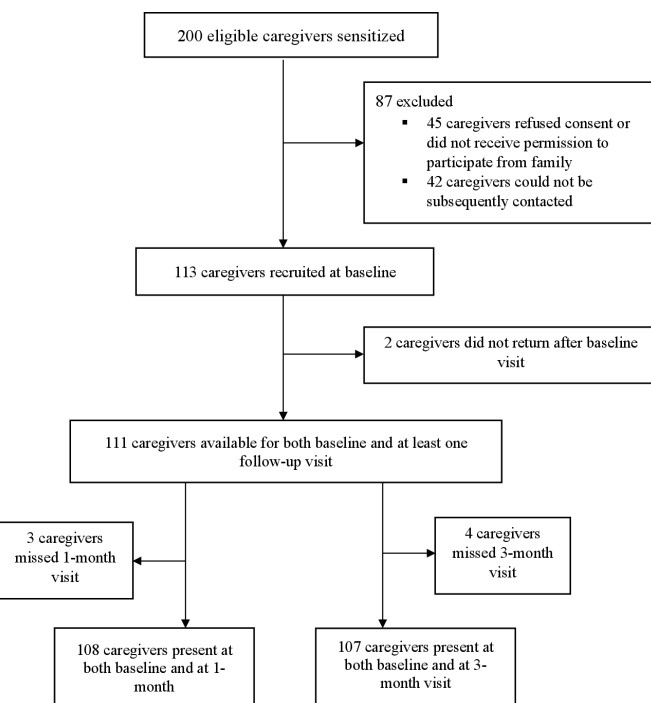

**Figure 2** Flow chart of participant recruitment process.

with significance level set at p<0.05. All quantitative data were entered into Research Data Capture for proper documentation and analyses were performed using Stata V.13 (StataCorp, USA).

For the qualitative analysis, we applied inductive thematic analysis in deriving our themes and subthemes as described by Braun and Clarke.[18] We used NVivo (V.12) software to organise the qualitative data during the analysis.

### Patient and public involvement
The public (caregivers and HCWs) were involved in the design of the SB and the questionnaires used for data collection tool.

### RESULTS
Between January and July 2019, we approached 200 eligible caregivers to participate in the study of which a total of 113 were enrolled after exclusion of 45 caregivers due to refusal of consent or absence of permission to participate from family, and 42 caregivers who could not be subsequently contacted on the phone numbers and addresses provided (figure 2). Of the 113 caregivers, 104 (92%) completed all three study visits, 108 (95.6%) completed the baseline and 1-month follow-up visits, and 107 (94.7%) completed the baseline and 3-month follow-up visits. Reasons for missed follow-up visits were ill-health of the mother, ill-health of infant and change in location.

All consenting caregivers were biological mothers of the infants presenting at the health facility. The mean age of the caregivers was 26.7±5.95 years. Most of the

caregivers were married (n=106, 93.8%), Muslim (n=111, 98.2%) and unemployed (n=91, 80.5%). Over two-thirds of the caregivers reported that they lived within 1 hour of the immunisation clinic, and most women (n=109, 96.5%) had received at least one dose of tetanus toxoid during their last pregnancy. Although none of the caregivers identified the media as their source of information regarding immunisation, 90 (79.6%) households had a functional radio, and about half of them had a television in their homes. Over two-thirds of the caregivers acknowledged that HCWs at the immunisation clinics were their major source of information on immunisation. Table 1 summarises the sociodemographic characteristics of the caregivers.

### Impact of SB on knowledge of caregivers
There was a significant increase in the median knowledge score from 6.0 (IQR 5.0–7.0) at baseline to 11.0 (IQR 8.0–14.0) at 1-month visit (p<0.001), and 15.0 (IQR 10.0–20.0) at 3-month visit (p<0.001). The highest median knowledge scores at 1-month and 3-month visits were seen among the caregivers above 30 years of age and families with monthly household income above $100 (13.0 and 17.0, respectively). When analysed by subgroups (urban or rural, age of caregivers, household income or highest level of education), there was significant improvement in knowledge scores across all groups, as summarised in table 2.

### Opinions of the caregivers
In general, the SB was very well received by the caregivers, and their opinions are summarised in table 3. The caregivers found the SB increasingly easy to use (103 of the 108 (95.4%) at 1 month vs 107 of the 107 (100%) at 3 months, respectively), and most of them stated that they understood all the information in the book (100 of the 108 (92.6%) at 1 month vs 98 of the 107 (91.6%) at 3 months). Many participants also shared the SB in their communities and expressed trust in the information provided. The survey results also showed that using the book was easy, with only 10 (9.3%) and 7 (6.5%) caregivers reporting some problems at 1-month and 3-month visits, respectively.

### Themes from qualitative questions posed to caregivers in questionnaire
In addition to the questions posed to caregivers on their experiences using the SB from the open-ended part of the questionnaire, a majority of caregivers (n=98, 90.7% and n=100, 93.5%) reported no problems while using the SBs at 1-month and 3-month post-visit, respectively. Reoccurring themes with supporting quotes are presented below:

#### Information appropriateness
I trust the messages in the book because the nurses tell me the same information as stated in the book. (32 years old, French language certificate, urban dweller)

| Table 1 Sociodemographic characteristics of caregivers enrolled | | |
|---|---|---|
| **Variable** | **N=113** | **%** |
| Age (years) | | |
| <20 | 23 | 20.4 |
| 21–30 | 64 | 56.6 |
| >30 | 26 | 23.0 |
| Religion | | |
| Muslim | 111 | 98.2 |
| Christian | 2 | 1.8 |
| Region | | |
| Urban | 52 | 46.0 |
| Rural | 61 | 54.0 |
| Marital status | | |
| Unmarried | 7 | 6.2 |
| Married | 106 | 93.8 |
| Employment status | | |
| Employed | 22 | 19.5 |
| Unemployed | 91 | 80.5 |
| Highest level of education | | |
| No education | 42 | 37.2 |
| Primary | 11 | 9.7 |
| Secondary | 21 | 18.6 |
| Higher secondary | 21 | 18.6 |
| Tertiary | 2 | 1.8 |
| Arabic/French | 16 | 14.2 |
| Monthly household income | | |
| Below $50 | 19 | 16.8 |
| $50–$100 | 38 | 33.6 |
| $100–$200 | 14 | 12.4 |
| Above $200 | 3 | 2.7 |
| Don't know | 39 | 34.5 |
| Presence of working radio in the home | | |
| Yes | 90 | 79.6 |
| No | 23 | 20.4 |
| Presence of working television in the home | | |
| Yes | 56 | 49.6 |
| No | 57 | 50.4 |
| Reported distance to health facility | | |
| Less than 30 min | 41 | 36.3 |
| 30 min–1 hour | 37 | 32.7 |
| Over 1 hour | 13 | 11.5 |
| Not sure | 22 | 19.5 |
| Mother's receipt of tetanus toxoid | | |
| Yes | 109 | 96.5 |
| No | 4 | 3.5 |
| Usual source of information regarding immunisation | | |

Continued

| Table 1 Continued | | |
|---|---|---|
| **Variable** | **N=113** | **%** |
| Health workers | 77 | 68.1 |
| Infant welfare card | 6 | 5.3 |
| Family member or friend | 4 | 3.5 |
| Media | 0 | 0.0 |
| Not sure | 26 | 23.0 |

### Clear presentation

… because the book is written in English… and also clearly translated in two local languages. (31 years old, tertiary education, urban dweller)

…the pictures make me believe the messages. (30 years old, no education, rural dweller)

### Evidence of vaccine effectiveness

… I trust the messages because some diseases we used to see…. [giving examples] have now drastically reduce. (37 years old, secondary (grade 7–9), rural dweller)

Of the small number of caregivers who reported problems while using the SBs at 1-month (n=10, 9.3%) and 3-month (n=7, 6.5%) follow-up visits, respectively, the reoccurring themes with supporting quotes are presented below:

### Understanding information

… I found it difficult understanding the book at first, but after some time I began to understand better. (32 years old, secondary (grade 7–9), urban dweller)

… because I am not educated, I found the book difficult to learn despite the audio… I began to understand with the help of some students who read and explained to me. (25 years old, secondary (grade 7–9), urban dweller)

### Sense of responsibility

… Initially I was afraid the book would go bad in my care but with time I was not afraid anymore. (27 years old, no education, rural dweller)

### Challenges with the batteries

… The battery died and then I had to change it. (25 years old, tertiary (university education), urban dweller)

### Themes and subthemes from IDIs with HCWs

IDIs were conducted with 14 of the 15 (93.3%) HCWs at the immunisation clinics. The mean age of the HCWs interviewed was 33.4 years, and they were predominantly women (64.3%). One of the HCWs could not be interviewed as they were absent from work due to ill-health. Themes and subthemes are summarised in figure 3. See also online supplemental material 5.

**Table 2** Median (IQR) of the knowledge scores at baseline, 1-month and 3-month visits*

| Variable | N=108 | Median baseline knowledge score† | Median knowledge score at 1 month† | Median difference† (1-month–baseline score) | P value‡ | N=107 | Median baseline knowledge score§ | Median knowledge score at 3 months§ | Median difference§ (3-month–baseline score) | P value‡ |
|---|---|---|---|---|---|---|---|---|---|---|
| All caregivers | 108 | 6.0 (5.0–7.0) | 11.0 (8.0–14.0) | 5.0 (2.0–8.0) | <0.0001 | 107 | 6.0 (5.0–7.0) | 15.0 (10.0–20.0) | 8.0 (5.0–14.0) | <0.0001 |
| Region | | | | | | | | | | |
| Rural | 57 | 6.0 (5.0–7.0) | 11.0 (8.0–15.0) | 4.0 (2.0–9.0) | <0.0001 | 58 | 6.0 (5.0–7.0) | 15.5 (10.0–21.0) | 10.0 (5.0–15.0) | <0.0001 |
| Urban | 51 | 6.0 (4.5–7.0) | 11.0 (7.0–14.0) | 5.0 (2.0–7.0) | <0.0001 | 49 | 6.0 (4.5–7.0) | 13.0 (11.0–19.0) | 8.0 (5.0–12.0) | <0.0001 |
| Age (years) | | | | | | | | | | |
| <20 | 23 | 6.0 (5.0–7.0) | 9.0 (8.0–14.0) | 4.0 (2.0–7.0) | 0.0001 | 23 | 6.0 (5.0–7.0) | 12.0 (8.0–19.0) | 6.0 (2.0–14.0) | <0.0001 |
| 21–30 | 60 | 6.0 (5.0–7.0) | 11.0 (7.0–14.0) | 4.0 (2.0–8.0) | <0.0001 | 61 | 6.0 (5.0–7.0) | 15.0 (11.0–21.0) | 8.0 (5.0–14.0) | <0.0001 |
| >30 | 25 | 6.0 (4.0–7.0) | 13.0 (10.0–16.0) | 6.0 (4.0–10.0) | <0.0001 | 23 | 6.0 (4.0–7.0) | 16.0 (12.0–17.0) | 9.0 (6.0–13.0) | <0.0001 |
| Household income | | | | | | | | | | |
| <$50 | 19 | 6.0 (5.0–7.0) | 11.0 (10.0–14.0) | 5.0 (4.0–8.0) | 0.0002 | 18 | 6.0 (5.0–7.0) | 15.5 (13.0–23.0) | 9.5 (6.0–17.0) | 0.0002 |
| $50–$100 | 38 | 6.0 (5.0–7.0) | 11.0 (8.0–14.0) | 5.0 (3.0–7.0) | <0.0001 | 37 | 6.0 (5.0–7.0) | 14.0 (11.0–20.0) | 8.0 (5.0–13.0) | <0.0001 |
| >$100 | 15 | 6.0 (5.0–7.0) | 12.0 (8.5–15.5) | 7.0 (2.5–9.5) | 0.0009 | 13 | 6.0 (5.0–7.0) | 17.0 (14.0–20.0) | 10.0 (7.0–13.0) | 0.0004 |
| Not sure | 36 | 6.0 (5.0–6.0) | 9.0 (7.0–15.0) | 3.5 (1.5–9.5) | <0.0001 | 39 | 6.0 (5.0–6.0) | 12.0 (8.0–20.0) | 7.0 (3.0–15.0) | <0.0001 |
| Level of education | | | | | | | | | | |
| None | 41 | 6.0 (5.0–7.0) | 11.0 (8.0–15.0) | 5.0 (2.0–10.0) | <0.0001 | 41 | 6.0 (5.0–7.0) | 15.0 (11.0–21.0) | 9.0 (5.0–15.0) | <0.0001 |
| Primary | 11 | 5.0 (4.0–6.0) | 10.5 (8.0–15.0) | 5.5 (4.0–9.0) | 0.0057 | 11 | 5.0 (4.0–6.0) | 16.5 (12.0–22.0) | 12.0 (8.0–17.0) | 0.005 |
| Secondary | 19 | 6.0 (5.0–7.0) | 11.0 (9.0–12.0) | 5.0 (3.0–7.0) | 0.0001 | 19 | 6.0 (5.0–7.0) | 13.0 (10.0–16.5) | 7.5 (5.0–10.0) | 0.0001 |
| Higher secondary/tertiary | 22 | 6.0 (5.0–7.0) | 10.5 (8.0–13.0) | 4.5 (2.0–8.0) | 0.0002 | 21 | 6.0 (5.0–7.0) | 16.5 (10.0–23.0) | 11.0 (5.0–16.0) | 0.0001 |
| Arabic/French | 15 | 6.5 (6.0–7.0) | 10.0 (7.0–15.0) | 3.0 (1.5–8.5) | 0.0048 | 15 | 6.0 (5.0–7.0) | 13.0 (8.0–17.5) | 6.0 (2.5–11.5) | 0.0009 |

*Since medians (IQR) of the differences are reported, these might not match the difference of the medians at each time point.
†Median baseline knowledge score and median difference calculated using the same participants who were present at both baseline visit and 1-month follow-up visit.
‡Wilcoxon signed-rank test p values.
§Median baseline knowledge score and median difference calculated using the same participants who were present at both baseline visit and 3-month follow-up visit.

**Table 3** Opinions of caregivers about the Speaking Book

| Questions | | 1-month visit | | 3-month visit | |
|---|---|---|---|---|---|
| | | n=108 | (%) | n=107 | (%) |
| Did you find the book easy to use? | Yes | 103 | 95.4 | 107 | 100 |
| | No | 5 | 4.6 | 0 | 0 |
| Did you like the pictures in the book? | Yes | 108 | 100 | 107 | 100 |
| | No | 0 | | 0 | 0 |
| Could you hear the person talking to you clearly? | Yes | 108 | 100 | 107 | 100 |
| | No | 0 | | 0 | 0 |
| Did you understand all the information that you were told in the book? | Yes | 100 | 92.6 | 98 | 91.6 |
| | No | 8 | 7.4 | 9 | 8.4 |
| Did you find the information in the book useful? | Yes | 107 | 99.1 | 106 | 99.1 |
| | No | 1 | 0.9 | 1 | 0.9 |
| Did you trust the messages in the book? | Yes | 108 | 100 | 106 | 99.1 |
| | No | 0 | 0 | 1 | 0.9 |
| Did you learn new information from the book? | Yes | 102 | 94.4 | 105 | 98.1 |
| | No | 6 | 5.6 | 2 | 1.9 |
| How many times did you go through the whole book? | Once | 4 | 3.7 | 5 | 4.7 |
| | 2–5 times | 40 | 37.0 | 33 | 30.8 |
| | 6–9 times | 10 | 9.3 | 3 | 2.8 |
| | ≥10 times | 54 | 50.0 | 66 | 61.7 |
| Can you clearly explain the information in the book to your family and friends? | Yes | 107 | 99.1 | 102 | 95.3 |
| | No | 1 | 1.0 | 5 | 4.7 |
| Do you think that the information in the book gives you all the information needed to decide to immunise your child? | Yes | 108 | 100 | 106 | 99.1 |
| | No | 0 | 0 | 1 | 0.9 |
| Were there any problems you had when you were using the book? | Yes | 10 | 9.3 | 7 | 6.5 |
| | No | 98 | 90.7 | 100 | 93.5 |
| Do you think the book should include any other information or have any changes? | Yes | 30 | 27.8 | 20 | 18.7 |
| | No | 78 | 72.2 | 87 | 81.3 |
| Did you show the book to anyone in your family? | Yes | 107 | 99.1 | 106 | 99.1 |
| | No | 1 | 1.0 | 1 | 0.9 |
| Did you show the book to anyone in your community? | Yes | 87 | 80.6 | 96 | 89.7 |
| | No | 17 | 15.7 | 11 | 10.3 |
| | No response | 4 | 3.7 | 0 | 0.0 |
| Did you show the book to anyone in your mosque/church? | Yes | 4 | 3.7 | 2 | 1.9 |
| | No | 104 | 96.3 | 104 | 97.2 |
| | No response | 0 | 0 | 1 | 0.9 |
| Did you show the book to anyone at work? | Yes | 21 | 19.4 | 16 | 15.0 |
| | No | 86 | 79.6 | 91 | 85.0 |
| | No response | 1 | 0.9 | 0 | 0 |
| Did you show the book to anyone at the clinic or hospital? | Yes | 10 | 9.3 | 14 | 13.1 |
| | No | 95 | 88.0 | 90 | 84.1 |
| | No response | 3 | 2.8 | 3 | 2.8 |
| Do you think members of your family, mosque/church and community will understand the information in the book if they were given this book to listen to? | Yes | 106 | 98.1 | 100 | 93.5 |
| | No | 0 | 0.0 | 1 | 0.9 |
| | I don't know | 2 | 1.9 | 6 | 5.6 |

**Table 3** Continued

| Questions | | 1-month visit | | 3-month visit | |
|---|---|---|---|---|---|
| | | n=108 | (%) | n=107 | (%) |
| At what time do you think that the books should be given to a parent coming to get their child immunised? | At the time of first visit to the clinic only | 82 | 75.9 | 102 | 95.3 |
| | At first few visits to the clinic | 11 | 10.2 | 5 | 4.7 |
| | While speaking to the HCW | 1 | 0.9 | 0 | 0 |
| | Before speaking to the HCW | 2 | 1.9 | 0 | 0 |
| | After speaking to the HCW | 10 | 9.3 | 0 | 0 |
| | No response | 2 | 1.9 | 0 | 0 |
| Do you know where the on/off switch is in the book? | Yes | 108 | 100 | 107 | 100 |
| | No | 0 | 0.0 | 0 | 0 |
| Would you be able to change the battery? | Yes | 95 | 88.0 | 106 | 99.1 |
| | No | 11 | 10.2 | 1 | 0.9 |
| | No response | 2 | 1.9 | 0 | 0 |
| If you need to change the battery, do you have a screwdriver with which to do so? | Yes | 83 | 76.9 | 97 | 90.7 |
| | No | 25 | 23.1 | 10 | 9.3 |

HCW, healthcare worker.

## Theme 1: general challenges at the clinics, independent of the SB
### Subtheme 1.1: delivering health education
From our discussions, we found out that HCWs were expected to deliver 'health talks' in the clinics before immunisation sessions. These health talks would typically include information about the vaccines their infants would receive that day, address the general expectations of caregivers and remind the caregivers when to return for their next appointment. In some cases, HCWs would add points that are important for the health of the babies like regular weighing, nutrition, among others.

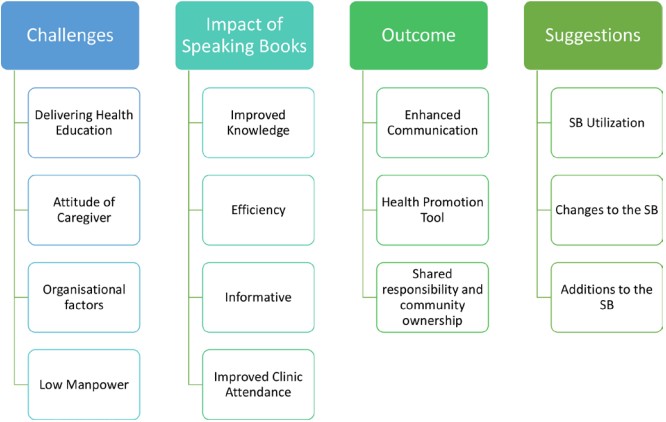

**Figure 3** Themes and subthemes from in-depth interviews with healthcare workers.

… we normally conduct pre-clinic health talks… We discuss vaccines, and sometimes we choose other topics… about their health and that of their babies. (Male, rural clinic, 18 months in current position)

Discussions with HCW suggested that these vaccine clinic health talks are not standardised, and the content and quality vary between locations and situations. Immunisation activities in facilities were concentrated in the morning hours of the day. While HCWs planned preclinic health talks at the beginning of every immunisation clinic, several of them ended up only briefly summarising the plan before vaccinating, while some resorted to giving caregivers briefs while administering vaccines at the immunisation table. Others did not bother to give the health talk at all due to time constraints. This was confirmed by one of the HCWs who said:

When we have a large crowd, we also engage them in interpersonal communication while administering the vaccines, we tell them about the vaccines been administered and the possible side effects of the vaccine. (Male, rural clinic, 6 months in current position)

### Subtheme 1.2: attitude of caregiver
Several HCWs expressed dissatisfaction regarding the attitude of caregivers. According to them, some caregivers did not show up on their visit appointments, others came late,

while others were impatient and always in a hurry to go home. HCWs perceived caregivers to value domestic chores and activities more than getting the best out of the immunisation clinic visit. This was confirmed by a HCW who said:

> When women come to the clinic, they are always in hurry to go back home to cook or going back to the garden, so most times they are in haste to go home. (Male, urban clinic, 5 years in current position)

In ensuring caregivers sat and listened to the vaccine health education, HCWs developed various means of ensuring they stayed to receive the information. According to one of the HCWs:

> What we do sometimes is to hold their cards and start weighing, and then when most of the women are around, we start giving the health education. (Male, rural clinic, 8 months in current position)

### Subtheme 1.3: organisational factors

Our discussions with HCWs revealed that although clinic schedules were planned in advance, there were a limited number of immunisation clinics in a month, especially in rural settings. According to a few HCWs, overcrowding and congestion during immunisation clinics was common as many of the caregivers visited at the same time on the limited number of days allotted to immunisation services.

> …most of them [caregivers] come around the same time and the health care worker might not have time to carefully explain in detail to the women … most times the environment is even very noisy. (Male, rural clinic, 3 years in current position)

The clinic setting was sometimes found to be very noisy and chaotic. According to one of the HCWs:

> …. some of the women will be standing and distracted by the cry of their children and may not even have the patience to listen to what we are saying to them. Interpersonal health education is usually better for us with the crowd (large numbers of caregivers). (Female, rural clinic, 3 years in current position)

Our discussions revealed that these challenges cut across facilities in both the urban and rural centres. The HCWs further emphasised the importance of the SB in filling the gaps of improving the knowledge of caregiver.

### Subtheme 1.4: low manpower

Findings revealed that some facilities did not have adequate HCWs/PHOs in the clinic to deliver immunisation services. Staff shortages were stressful for both HCW and caregiver, resulting in long waiting time, poor service delivery and dissatisfaction. This was confirmed by one of the HCWs who said:

> … when we started the implementation of the speaking books, I was the only Public Health Officer here without any form of assistance or back up. (Male, urban clinic, 5 years in current position)

Another HCW said:

> Sometimes I will be the only one immunizing, weighing and screening so sometimes it's difficult to use the speaking books. (Male, urban clinic, 5 years in current position)

Staff shortages are stressful for both HCWs and caregivers, resulting in long waiting time, poor service delivery and dissatisfaction.

### Theme 2: SB impact
### Subtheme 2.1: improved knowledge

HCWs revealed that they observed increased knowledge among caregivers as a result of the utilisation of the SB. They added that the SB addresses vaccine issues adequately.

> Most times after listening to the books, we ask them questions on immunisation and they are able to give positive feedback. (Male, urban clinic, 18 months in current position)

> … I cannot assure you on their [all caregivers attending the immunisation clinic] increase in knowledge compared to women who have personal copies [recruited participants] with them at home. (Male, rural clinic, 8 months in current position)

### Subtheme 2.2: efficiency

Over the course of the discussion, HCWs expressed ways the SB has increased productivity with their work. Some of the HCWs emphasised the importance of the local languages used in the book. The SB was judged to be a self-explanatory tool which most of the caregiver could use with little or no assistance.

> … for me it saves time because I just hand over the book to them (caregivers) and then they listen to everything… while I concentrate on the work and it makes work faster. (Female, rural clinic, 3 years in current position)

Some of the HCWs expressed their desire to have more copies of the SB in the facilities to further increase productivity with their work.

> … the issue is we have just one copy in the facility which… is not enough… they (caregivers) all have to keep waiting for each other to listen to the books… if we have at least five copies it would be better than everyone sharing one copy. (Male, urban clinic, 18 months in current position)

### Subtheme 2.3: informative

Some of the HCWs described the SB as an informative tool which served the needs of caregivers and HCWs. The content was able to address most of the concerns of caregivers and presented vaccine information in a simple way that could be easily understood.

> In the past they give so many complaints about adverse effects following vaccination……. even blame public health officers that administer the vaccines,

they have a better understanding of the side effects after receiving the vaccines and appreciate the information gotten from the book. (Female, rural clinic, 3 years in current position)

It has addressed all the details and relevant information that will make mothers want to bring their children for immunisation. (Female, rural clinic, 3 years in current position)

### Subtheme 2.4: improved clinic attendance

HCWs repeatedly highlighted enhanced clinic attendance by caregivers since the introduction of the SB, especially among those with personal copies of the SB. Caregivers appeared more motivated and dedicated to the immunisation schedule. Some HCWs suggested that the SB had resulted in increased uptake of vaccination, helping them to meet their coverage targets.

Mothers are beginning to bring their babies to the clinic unlike before when they will just come to the clinic anytime they like. (Female, urban clinic, 10 years in current position)

In the past we used to have low coverages but with the introduction of the speaking books are now seeing more mothers coming. (Male, urban clinic, 5 years in current position)

### Theme 3: outcome of use of the SB
### Subtheme 3.1: enhanced communication

Discussions focusing on how the SB had improved communication between HCWs and caregivers revealed that the two SB local languages had aided the sharing of vaccine information. Some HCWs expressed their struggle in explaining vaccine preventable diseases in a way that could be understood by mothers prior to the introduction of the SB.

The book has made it easier for us to explain information on vaccines to the mothers… (Female, urban clinic, 10 years in current position)

… most of the women don't understand English but with the use of the local languages they understand the information easily… It has been easier for us to pass across the information because English is not our language, so it is easier for them to understand and easier for us to explain to them. (Male, urban clinic, 3 years in current position)

### Subtheme 3.2: health promotion tool

As the discussions continued, some of the HCWs highlighted the importance of the SB to them and their colleagues in other facilities. They considered the SB an important tool for HCWs in general. They discussed the importance of sharing the information in the SB with HCWs in other facilities, including through recoding the audio on a phone and sharing with colleagues on social media. One HCW said:

… we are trying, by recording the audio on our phones and sharing with other health staff because

not all health officers have access to the information in this book… We learnt a lot from it. (Male, urban clinic, 3 years in current position)

Another HCW expressing his own unique way of making use of the SB said:

Because the book is not loud for all the mothers to hear, I recorded it on my phone, used a Bluetooth device to connect to a speaker for all the mothers to hear. (Male, urban clinic, 2 years in current position)

### Subtheme 3.3: shared responsibility and community ownership

According to the HCWs, some of the women shared information from the SB in the community and some women had been eager to take the responsibility of health talks during immunisation clinics with their peers. HCWs considered this as a way of spreading vaccine-related information and facilitating their own work.

They share during lunch or when they sit together, they play the book. (Male, rural clinic, 8 months in current position)

During the immunisation clinic days most of the women who have copies of the speaking books come along with the books and they normally take over the health talks with the use of their speaking books, sometimes I don't even play my own. (Male, urban clinic, 5 years in current position)

Further discussion revealed that caregivers made use of the SB in family and community gathering.

… in the communities, mothers and caregivers look up to those who have benefitted from the study they are like mentors to other mothers and caregivers in the community where they live. (Male, rural clinic, 3 years in current position)

### Theme 4: further suggestions
### Subtheme 4.1: SB utilisation

The opinion of HCWs regarding their preferences on the best setting for the utilisation of SB varied. Some HCWs did not make use of the SB during very busy clinics but instead preferred to use it when the clinic was quieter. Some suggested the SB to be more useful in homes with caregivers as they are able to go through the book at their own convenient time with the added advantage of sharing the SB with people in the communities who may not come to the clinic. They believed this would foster transfer of knowledge to the community members.

I think is better they are given the books at their homes… in the clinic most times they are in a hurry to go home… at home they will take out time to listen to the book and gain more information compared to listening to the books in the clinic. (Male, urban clinic, 5 years in current position)

…. I believe the women will make better use of the speaking books if they are used in the facilities than

in their homes … at the health facility the health workers will be around to guide them, this will ensure a better use of the book… (Male, urban clinic, 18 months in current position)

### Subtheme 4.2: changes to the SB

The low sound volume of the SB remained a consistent challenge expressed by the HCWs during the discussions; some even suggested modifying the SB so it could be connected to external devices that could amplify the volume.

The only problem the speaking book has is that it cannot be used with a large audience because the volume is very low. (Female, rural clinic, 3 years in current position)

…. it will be good to have a book that is very loud so we don't have to talk, we will just place it at the front of the Clinic, press the buttons and the book will start playing for the women to listen to it. (Male, rural clinic, 6 months in current position)

### Subtheme 4.3: additions to the SB

Constraints with optimal utilisation of the SB were linked to the use of only two languages in the audio translation of the book. In addition to the challenges faced in recruiting caregivers for this study, HCWs further suggested the need to incorporate additional languages to make use of the book with more caregivers of different language backgrounds.

… other languages need to be added…. because caregivers who do not understand any of the languages used in the book did not benefit from the information in the book… (Male, urban clinic, 4 years in current position)

Some HCWs suggested other health topics which could be added to the vaccine SB, such as nutrition and maternal health.

…. we can also consider adding information on nutrition and importance of antenatal care by pregnant women. (Female, urban clinic, 10 years in current position)

## DISCUSSION

In this study, we assessed the impact of a bespoke, multimedia educational tool, the vaccine SB, on vaccine-related knowledge of caregivers as well as on communication between HCWs and caregivers in The Gambia. We distributed this tool to 113 caregivers visiting PHC for routine immunisation services and evaluated changes in vaccine knowledge over a 3-month period.

We saw significant improvement in the knowledge scores with the median scores nearly doubling at 1-month and nearly tripling at 3-month follow-up visits, compared with the baseline scores, independent of region of the country (urban or rural), age of caregiver, household income or highest level of education. The SB was given to the caregivers to take home, thus giving them several opportunities to listen to and understand the information in the book, which might have contributed to the ongoing improvement over the 3-month period.

Study participants discussed and shared content of the SB extensively within their family, community and social groups, using a variety of platforms such as peer group meetings, community and religious gatherings. This is likely to lead to a significant and possibly measurable multiplier effect. Our results support the usefulness of the SB beyond individuals' knowledge gain and in both the health facility and community setting. Our findings of 'added value' are similar to observations by other investigators who used a similar tool.[17 19] One study in South Africa reflected the use of an SB tool as significant in increasing knowledge on clinical trials among recruited study participants.[17] Another study reflected the effectiveness of the tool in increasing knowledge of biobanking and genetics among a group of non-academic university staff in South Africa. This qualitative experimental study reported a significant increase overall in knowledge score, similar to our findings, although this increase in knowledge score did not occur for all questions.[17 19]

Our results support the importance of providing targeted, vaccine-related education to caregivers and communities as an effective and practical strategy to improve vaccine-related knowledge in low-literacy settings such as The Gambia.[20 21]

Our data are comparable with findings from a previous study conducted by our team in The Gambia[22] where the use of the SB equally resulted in significant improvement in knowledge gained among study participants. However, the previous SB was directed at providing information about a specific clinical trial of pneumococcal vaccines, not the EPI programme and vaccine-related knowledge in general. It was also conducted in a community with previous exposure to vaccine research, a setting we deliberately avoided in the project presented here as it is likely to lead to significant bias. Illustrations in the previous SB had also not been adapted to the local Gambia context. During the stakeholder engagement for our study, we received feedback that led to specific alterations with regard to dress code, for example, and to derive an SB that reflects local communities, which was an important message received.

While the SB was able to significantly improve the vaccine-related knowledge, a knowledge gap remained: although we reported significant improvements in the mean knowledge scores from a very low baseline, the highest score obtained was 19.0, still 20 points below the maximum possible score of 39. It is possible that the detailed knowledge needed to obtain these remaining points is more difficult to retain compared with some other answers and we wondered if there was an educational bias. The subgroup of caregivers with tertiary education did not achieve significantly higher

scores, maybe because they already had a higher baseline score. We are unable to draw firmer conclusions about the impact of higher education, since there were only two participants in this group. It is entirely possible that the knowledge score developed was overambitious.

A majority of the caregivers reported that HCWs were their usual source of information about vaccines. Hence, further training and retraining the HCWs to deliver the additional information to the caregivers could significantly contribute to reducing the remaining knowledge gap.[23] In a study similar to ours, the SB was used regularly for health promotion in health posts, home visits and local gatherings and HCWs referred to the SB as a 'job aid'.[15] They carried out observations, structured interviews and FGDs with HCWs within a selected region of Ethiopia, this informed their reports on the contribution, appropriateness and challenges of the SB.[15]

Most caregivers had listened to the SB multiple times and suggested that it would be important for mothers to receive a copy of the SB at their first clinic visit, especially while pregnant or when their babies are newly born. Studies have reported distribution of the SB to study participants who have either had the opportunity to spend days with the tool, use the tool at workstations or take the tool home as their personal property.[15 22] Trust in the messages provided by the book was almost unanimous and the illustrations, which were adapted to the local context, were approved by all participants. The initial stakeholder consultation in the design phase of the book was therefore very important. We highly recommend to involve the local communities in the design of such tools, as their identification with the visual materials is likely to increase its acceptancy.[12 16]

Results obtained from the HCW interviews demonstrated that the SB had empowered them to answer concerns and questions of caregivers, emphasising the importance of the local language translations which presented vaccine information in a simple, easy-to-understand manner. HCWs reported increase in productivity and improved clinic attendance and considered the SB as an important health promotion tool for both HCWs and caregivers in The Gambia. It was evident that the SB had a positive impact on their day-to-day practice, and its content was well received, which facilitated communication on vaccines.

Vaccine confidence remains high in countries like The Gambia, but in the overall context of increasing vaccine hesitancy worldwide, tools such as the SB or similar approaches tailored to local context might add significant value to engage individuals and communities in communication about the personal and societal value of vaccination. They could be an important countermeasure to vaccine hesitancy.[24–26] It is interesting to note that none of the participants reported obtaining their vaccine-related information from the media, unlike in many HICs.

We acknowledge the lack of a control group in our study, but this was beyond our means in this project. We agree that the inclusion of a control group not receiving the SB would have been desirable but funding restrictions did not allow us to implement this design as we needed to make sure we had a large enough sample of caregivers who actually received the SB and were also representative of the country. This included urban and rural areas and a different set-up of health clinics. Further evaluations could be carried out in a properly designed and powered trial in the future and we believe that our study provides useful data for the conduct of such a study.

Overall vaccine literacy is more complex than 'just' being knowledgeable about schedules and target diseases of the vaccines provided in the EPI schedule, which was the focus of most of our knowledge test. It encompasses understanding, appraisal and application of vaccine-related information services, additional components not captured in this project.

Further research will now be required to quantify any potential effect on vaccine uptake at the health centre level and measure cost-effectiveness. In discussions with the Gambian EPI programme, both are expressed areas of interest. The SB used in this study was only recorded in two of the locally spoken languages (Wolof and Mandinka) in The Gambia and therefore, excluded caregivers who did not speak or understand either of these languages. We were asked to produce books in additional local languages in the future as well as design books for additional topics, such as pregnancy and nutrition.

While useful and possibly a first step to behaviour change, the intervention (ie, the SB) can of course not address multiple other barriers families face in getting their children vaccinated or the health system barriers' such as lack of staff, identified by the HCWs.

## CONCLUSION

In conclusion, the vaccine SB is a health education tool that resulted in significant increase in vaccine-related knowledge of caregivers, irrespective of their age, location, household income or educational level. It also served as a valuable tool for HCWs in their interactions with caregivers and might lead to increased uptake of immunisation services.

**Author affiliations**
[1]Vaccines and Immunity Theme, Medical Research Council Unit The Gambia at the London School of Hygiene and Tropical Medicine, Banjul, The Gambia
[2]Global Health, Global Healthcare Consulting, New Delhi, India
[3]Global Health, University of Washington Department of Global Health, Seattle, Washington, USA
[4]Expanded Programme on Immunization, Ministry of Health and Social Welfare, The Gambia, Banjul, The Gambia
[5]The Vaccine Centre, London School of Hygiene and Tropical Medicine, London, UK

**Acknowledgements**  We thank the field assistants for detailed data collection and transcription from the local languages to English for data analysis. We thank Alexander Jarde for statistical support in the analysis of the study data. We also thank the heads of the participating health facilities and members of staff for granting us access to the EPI clinics and their support throughout the project.

**Contributors**  BK and SK conceptualised and designed the study. OON, OW and AC were involved in data collection and analysed the data. OON wrote the first draft. BK, SK, OW, PJ and MK provided critical feedback and helped shape the research, analysis and manuscript. All the authors revised the work for important intellectual content and contributed to the final draft and finally approved it to be published.

**Funding**  This study was funded by a project grant from Bull City Learning. BK is supported by funding from the UKRI (MC_UP_A900/1122). The Vaccines and Immunity research led by BK is jointly funded by the UK Medical Research Council (MRC) and the UK Department for International Development (DFID) under the MRC/DFID Concordat agreement and is also part of the EDCTP2 programme supported by the European Union.

**Competing interests**  None declared.

**Patient consent for publication**  Not required.

**Ethics approval**  Ethical approval was obtained from The Gambia Government/MRC Joint Ethics Committee (SCC 1598).

**Provenance and peer review**  Not commissioned; externally peer reviewed.

**Data availability statement**  The data sets generated and analysed during this study are available from the corresponding author upon reasonable request.

**ORCID iDs**
Oluwatosin O Nkereuwem http://orcid.org/0000-0001-6194-6228
Oghenebrume Wariri http://orcid.org/0000-0002-7432-8995
Beate Kampmann http://orcid.org/0000-0002-6546-4709

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
