## [Reviewer comments · BMJ Open]

ARTICLE DETAILS

TITLE (PROVISIONAL)	THE USE OF A SPEAKING BOOK® TO ENHANCE VACCINE KNOWLEDGE AMONG CAREGIVERS IN THE GAMBIA
AUTHORS	Nkereuwem, Oluwatosin; Kochhar, Sonali; Wariri, Oghenebrume; Johm, Penda; Ceesay, Amie; Kinteh, Mamanding; Kampmann, Beate

VERSION 1 – REVIEW

REVIEWER	Yu-Chih Chen The University of Hong Kong
REVIEW RETURNED	22-Jun-2020

GENERAL COMMENTS	This study evaluates the effectiveness of Speaking Book (SB) on vaccine knowledge among caregivers in The Gambia using mix-method approach. I think this study is well-conceived and the paper is clear and well-written. My comments are minor and they can be addressed either by adding more details or discussed in the limitation section. First, as this study does not involve control group, as discussed in limitation, this study may not be free from self-selection bias and this should be addressed in text, with recommendation for how to handle such a bias. Second, as this study collects immunization history, it will be great if authors can present this information, as prior immunization experience may also influence the propensity of vaccination. Lastly, it is great to see authors uses triangulation by combining both quantitative (on caregivers) and qualitative (on health care workers) methods to present the effectiveness of SB. I will be interested in why authors do not conduct either focus group or in-depth interview on caregivers, but on health care workers specifically, in understanding about the perception of acceptability, efficacy, or challenges of vaccination. A typo in Table 3 should be corrected. The percentage for 1-month visit (line 48) should be 100(%), not 0.
--

REVIEWER	Ilaria Montagni Bordeaux Population Health UMRS1219 - University of Bordeaux - Inserm
REVIEW RETURNED	21-Jul-2020

GENERAL COMMENTS	The paper describes the development and evaluation of a multi-media educational tool to improve vaccine knowledge among Gambian caregivers and healthcare workers. The impact of the tool was assessed through a sequential mixed-methods approach with repeated measures (one month and three months after receiving the tool). The importance of the context and the
--

	specificity of the subject are recognized as important elements for the effectiveness of the tool which was appreciated by users. Limitations of the study are well reported, i.e. lack of a randomized trial approach (control group) and systematic measure of the impact through data from immunization services. The paper is well-written and the contents are clear. However, some details are missing before full publication.  - The title first sentence “Getting the message across” is not really useful to understand the content of the paper. It can be deleted to gain some characters. - In the introduction, it might be mentioned the fact that one of the causes of sub-optimal immunization in low-and-middle income countries can be vaccine hesitancy, which is not explicitly mentioned in the article. Hesitancy might be due to distrust in medical solutions compared to other traditional sources of care, for instance. You just rapidly mention this in the discussion, please expand. - The concept of vaccine literacy is also not explicitly reported, even if it is pivotal. It is not simply a matter of knowledge about vaccines, but of access, understanding, appraisal and application of vaccine-related information. - The EPI acronym must be explained the first time it is mentioned, see section “Study design and setting”. - The number of pre-testers of the questionnaire should be reported. - Please provide the exact number of closed and open items of the final questionnaire (not only the knowledge-related ones) and, if possible, provide the questionnaire as Supplemental Material. It is of particular interest to look at all questions on knowledge, especially to understand the low scores compared to the maximum of 39 points. - Similarly, provide more details about the interview grid: how many questions/themes/subthemes? Were these predefined or not? Why do you present recurring themes only in the results? Is this because they were identified a posteriori? - Figure 3 could be reorganized in a more structured grid, i.e. the real tool you used for performing the interviews. - The description of the Study Tool should precede the description of the methods (quantitative and qualitative) to evaluate it. - Please specify in the text of Methods that Figure 1 just show some excerpts. Complementing the excerpts, a picture of the real SB might be useful to fully understand the format of this type of tool (not intuitive when first reading). - Please specify why the illustrations were culturally sensitive. - In the Data Analysis, specify how you defined the sub-groups (this information must be put here and not exclusively in the results section). - You say that 113 caregivers agreed to participate, but in the flowchart the difference between those who refused consent and those who could not be contacted is not clear. Were those who refused to participate unable to read and write? This might be an explanation for not responding the questionnaires (if self-administered). - Is it implicit that all caregivers are only women? It should be clearly stated. Were all caregivers mothers of concerned children? - In Table 2, what p values stand for is not easy to read? Are they referring to differences between medians or across groups? Differences across groups would be more relevant. All differences between medians are statistically significant.
--	--

	 - Can you report which were the problems some caregivers encountered when using the SB? - In Table 3 there is an error in the item “Do you think that the information in the book gives you all the information needed to decide to immunize your child?” since the percentage 100 is not reported (instead of “0”). - Was the questionnaire self-administered? If not, why are there some missing data? - Was the questionnaire in paper format? How where the answers coded? Manually and by two coders or just one? How to avoid transcription errors? - The verbatim of the interviews with the HCW are very useful but they could be better structured in separate tables, if possible (see limited number of tables accepted by the journal, one box composed of the different themes could be a good solution). - You mention in the Discussion two studies reporting “added value” of tools like the SB. Contextualize these studies (where, with whom?) to better appraise comparability with your work.
--	--

REVIEWER	Jo Durham Queensland University of Technology, Australia
REVIEW RETURNED	01-Aug-2020

GENERAL COMMENTS	This is an interesting and well written paper. Some brief information about the EPI program and vaccinations rates in the regions included, literacy rates etc. in the study would be helpful to provide context Also make the purpose of the paper study either – overall what is the purpose your argument? What do you want to persuade the reader? How does this work fit into the broader literature and do we learn? More information needs to be provided on o how the visual SB and how it was developed. Also why was the written language in English? And what does “context” refer to – is it just visual context in terms of villages/dress/facial feature etc for example or does it contain local understandings of vaccinations and misconceptions? In the findings I feel the qualitative findings need to be developed more rather than just the name of the theme and a quote. They should be presented with narrative and some interpretation The discussion could be more concise and focus more on interpretation rather than re-reporting the findings. Why do you think the SB worked/didn’t work? Why do you think participants had a high level of trust in the messages? Please check all references have all correct information
--

VERSION 1 – AUTHOR RESPONSE

Reviewer 1	Authors response
This study evaluates the effectiveness of Speaking Book (SB) on vaccine knowledge among caregivers in The Gambia using mix-method approach. I think this study is well-conceived and the paper is clear and well-written. My comments are minor, and they can be	We thank the reviewer for these very positive and helpful comments. We have revised the manuscript to address the comments and provide below a point-by-point response to each of the specific comments.

addressed either by adding more details or discussed in the limitation section	
First, as this study does not involve control group, as discussed in limitation, this study may not be free from self-selection bias and this should be addressed in text, with recommendation for how to handle such a bias.	We agree that the inclusion of a control group not receiving the SB would have been desirable but funding restrictions did not allow us to implement this design as we needed to make sure we had a large enough sample of caregivers who actually received the SB and were also representative of the country. This included urban and rural areas and a different set up of health clinics. We have included a comment in the limitations of our study section in the discussion. Further evaluations could be carried out in a properly designed and powered trial in the future and we believe that our study provides useful data for the conduct of such a study. (Page 23, paragraph 1)
Second, as this study collects immunization history, it will be great if authors can present this information, as prior immunization experience may also influence the propensity of vaccination.	Minimal immunisation history was collected, for example on the tetanus coverage in pregnancy experienced by the participating women, and this is reported in the results section. Given that all women approached came into the study via the immunisation clinics and all had children this implies that they had been exposed to these services, but we did not collect more detailed vaccination histories in this project. We believe that a larger dataset might have to be collected to really account for the level of granularity that might shed light on the hypothesis that prior experience with the immunisation services might have shaped the perception or knowledge of the women who participated. This is an interesting hypothesis which we could not examine within this study. We have acknowledged this point in the discussion. All participants were recruited through the vaccine clinics hence had been exposed to vaccinations in one way or another. The interviews with health care workers further reflect on the challenges faced prior to the introduction of the SB and its impact since the introduction.

Lastly, it is great to see authors uses triangulation by combining both quantitative (on caregivers) and qualitative (on health care workers) methods to present the effectiveness of SB. I will be interested in why authors do not conduct either focus group or in-depth interview on caregivers, but on health care workers specifically, in understanding about the perception of acceptability, efficacy, or challenges of vaccination.	During the design phase of the SB, we carried out focus group discussions with healthcare workers and care givers alike, which formed an important aspect in the development of the Speaking Book. The primary goal of this study however was to assess if the use of an educational tool would enhance knowledge, understanding and recall of key vaccine-related information among caregivers in The Gambia and this was assessed using a quantitative approach. Although we did not conduct detailed focus groups in each health centre, the use of an interview administered questionnaire comprising of both open and closed ended questions gave us insights into the usefulness and acceptability of the Speaking Book as detailed focus groups also including the caregivers at the different health centres was beyond our means. We instead opted for in-depth interviews to assess acceptability and relevance of the Speaking Book among healthcare workers. We have acknowledged these potential limitations in the discussion, but we believe that our approach gathered both the primary quantitative data as well as qualitative insights from caregivers and healthcare workers.
A typo in Table 3 should be corrected. The percentage for 1-month visit (line 48) should be 100(%), not 0.	This error has been corrected.

Reviewer 2	Authors response
The paper describes the development and evaluation of a multi-media educational tool to improve vaccine knowledge among Gambian caregivers and healthcare workers. The impact of the tool was assessed through a sequential mixed-methods approach with repeated measures (one month and three months after receiving the tool). The importance of the context and the specificity of the subject are recognized as important elements for the effectiveness of the tool which was appreciated by users. Limitations of the study are well reported, i.e. lack of a randomized trial approach (control group) and systematic measure of the impact through data from immunization services. The paper is well-written, and the contents are clear. However, some details are missing before full publication.	We acknowledge with thanks the reviewer's comments.

The title first sentence “Getting the message across” is not really useful to understand the content of the paper. It can be deleted to gain some characters.	The title of the paper has been changed to reflect your suggestion. It now reads: “A sequential mixed-method study on the use of a Speaking Book® to enhance vaccine knowledge among caregivers in The Gambia.”
In the introduction, it might be mentioned the fact that one of the causes of sub-optimal immunization in low-and-middle income countries can be vaccine hesitancy, which is not explicitly mentioned in the article. Hesitancy might be due to distrust in medical solutions compared to other traditional sources of care, for instance. You just rapidly mention this in the discussion, please expand.	We acknowledge the importance of the topic and have added a sentence regarding vaccine confidence in the introduction, but vaccine confidence or hesitancy was not the focus of this work, which was to assess a tool to enhance vaccine knowledge. (Pages 5, paragraph 2, line 2)
The concept of vaccine literacy is also not explicitly reported, even if it is pivotal. It is not simply a matter of knowledge about vaccines, but of access, understanding, appraisal and application of vaccine-related information.	We of course agree with the statement made by the reviewer and it is indeed a wide array of factors. We however also believe that vaccine knowledge is a first and important step to mitigate some of these factors and to assess vaccine knowledge (as a part of vaccine literacy) was at the heart of this study. There is currently no evidence on the status of vaccine knowledge in the region. Our results clearly show low vaccine knowledge among recruited participants. We agree that there is a need to further explore access, appraisal and application of vaccine-related information and we have added a sentence in the discussion to reflect on these additional factors. (Page 23, paragraph 2)
The EPI acronym must be explained the first time it is mentioned, see section “Study design and setting”.	We agree with the reviewer and we have revised the text and made this correction.
The number of pre-testers of the questionnaire should be reported.	We have added the number of pre-tested participants (25 caregivers and 5 healthcare workers). (Page 6, paragraph 2)
Please provide the exact number of closed and open items of the final questionnaire (not only the knowledge-related ones) and, if possible, provide the questionnaire as Supplemental Material. It is of particular interest to look at all questions on knowledge, especially to understand the low scores compared to the maximum of 39 points.	We have provided the exact numbers of the closed and open-ended questions. We have also provided the questionnaire as part of the Supplemental Material. (Pages 6, paragraph 2)
Similarly, provide more details about the interview grid: how many questions/themes/subthemes? Were these predefined or not? Why do you present recurring themes only in the results? Is this because they were identified posteriori?	We have provided sample questionnaires and interview guide. Samples can be found in the supplementary appendix. Themes and subthemes were not predefined they were purely from the data collected during the interviews.

Figure 3 could be reorganized in a more structured grid, i.e. the real tool you used for performing the interviews.	We have re-organised the original Fig 3 into a more comprehensive table and provided the interview tool in the supplementary methods.
The description of the Study Tool should precede the description of the methods (quantitative and qualitative) to evaluate it.	We have made changes to this arrangement on our manuscript following your suggestion.
Please specify in the text of Methods that Figure 1 just show some excerpts. Complementing the excerpts, a picture of the real SB might be useful to fully understand the format of this type of tool (not intuitive when first reading).	We have added more pictures of the speaking book to Figure 1 and have revised the statement to reflect this.
Please specify why the illustrations were culturally sensitive.	The SB are produced by a commercial company and versions on other topics were available to us. These versions had drawings from a South African artist, depicting sceneries which were more fitting for South Africa. Following focus group discussions (FGD) with caregivers and healthcare workers prior to the production of the Vaccine SB, the FGD participants suggested that the SB would be more acceptable if books were developed to depict the Gambian setting. Local artists were engaged to assist with drawings to help in the production of the book. All voice overs were done by local media personnel who could speak the local languages well. This improved the acceptability of the tool by the communities.
In the Data Analysis, specify how you defined the sub-groups (this information must be put here and not exclusively in the results section).	We defined sub-groups by Region, Age (years), Household income and Level of education. This information is now included in the Data analysis section. (Page 7, paragraph 4, line 4).
You say that 113 caregivers agreed to participate, but in the flowchart the difference between those who refused consent and those who could not be contacted is not clear. Were those who refused to participate unable to read and write? This might be an explanation for not responding the questionnaires (if self-administered).	200 eligible caregivers were sensitized at the health facilities. However, only 113 who returned their completed consent forms were enrolled into the study. Of the 87 who did not, 45 of them either refused consent or did not receive the permission of their family to participate, while 42 of them could not be subsequently contacted on the phone numbers which they provided. The questionnaires were interviewer-administered, and therefore would not depend on the literacy (or lack of it) of the participants.
Is it implicit that all caregivers are only women? It should be clearly stated. Were all care-givers mothers of concerned children?	We have stated in our results section that “All consenting caregivers were biological mothers of the infants.” (Page 8, paragraph 2, line 1)
In Table 2, what p values stand for is not easy to read? Are they referring to differences between medians or across groups? Differences across groups would be more relevant. All differences between medians are statistically significant.	The P-values represent differences in the median knowledge score at two timepoints (between baseline and 1-month, and between baseline and 3-month visits). We used the Wilcoxon signed-rank test to assess whether the population median ranks differ

	significantly at the three time points i.e two dependent samples were selected from populations having the same distribution.
Can you report which were the problems some caregivers encountered when using the SB?	We discussed this in the manuscript under the points from open ended questions we asked the caregivers. Survey results showed that utilization of the book did not prove difficult. However, of the 10 (9.3%) and 7 (6.5%) caregivers who reported problems while using the speaking books at 1-month post visit and 3-months post visit respectively, recurring themes with supporting quotes were presented. (Page 15, paragraph 1)
In Table 3 there is an error in the item “Do you think that the information in the book gives you all the information needed to decide to immunize your child?” since the percentage 100 is not reported (instead of “0”).	We have corrected this error.
Was the questionnaire self-administered? If not, why are there some missing data?	The questionnaire was face-to-face interviewer administered. As discussed in the manuscript, two participants did not return after the baseline study, three participants missed the 1-month visit, and four participants missed the 3-month visit. This explains the missing information in our data.
The verbatim of the interviews with the HCW are very useful but they could be better structured in separate tables, if possible (see limited number of tables accepted by the journal, one box composed of the different themes could be a good solution).	Thank you for the helpful suggestion. We have replaced Figure 3 with a more detailed table in the appendix showing themes, sub themes and supporting quotes.
You mention in the Discussion two studies reporting “added value” of tools like the SB. Contextualize these studies (where, with whom?) to better appraise comparability with your work.	We have reflected more on this point for better appraisal of its comparability with our work. (Pages 22-23)

Reviewer 3	Authors response
This is an interesting and well written paper.	We acknowledge with thanks the reviewer’s appreciation of our work and provide below a point-by-point summary in response to each comment.
Some brief information about the EPI program and vaccinations rates in the regions included, literacy rates etc. in the study would be helpful to provide context.	We have added some information in the introduction which highlights data from the Gambia EPI. It reads as follows: “The Gambian expanded programme on immunization (EPI) programme is considered as highly successful compared to other countries in sub-Saharan Africa. The programme has consistently maintained coverage of the third dose of Diphtheria-Tetanus-Pertussis

	(DTP3) above 95% and DTP1 to DTP3 dropout rates below 10% since 2005.” (Page 4, paragraph 4)
Also make the purpose of the paper study either – overall what is the purpose your argument? What do you want to persuade the reader? How does this work fit into the broader literature and do we learn?	We acknowledge the reviewer’s comment. We believe that we have addressed specific questions which also address this comment in the answers to the reviewers above. We believe that we have clearly stated our goals, described the methods, and acknowledged limitations.
More information needs to be provided on how the visual SB and how it was developed. Also why was the written language in English? And what does “context” refer to – is it just visual context in terms of villages/dress/facial feature etc for example or does it contain local understandings of vaccinations and misconceptions?	We have given detail re: the development of the SB in the methods and referred to the existing literature of the purpose and previous use of SB. The written language is English because neither of the local languages in The Gambia are written down. The short-written information is in English as that is the main written language taught at school in the Gambia, sometimes Arabic. All healthcare workers in The Gambia are able to read and write English, hence this is the appropriate language to facilitate the initial communication via the SB. However, literacy amongst caregivers can be limited, hence the voice overs are the key features of the SB and the detail provided in the recordings goes beyond the short information in English. We have addressed the question re: local context in our reply to reviewer 2. The SB are made by a commercial company and versions on other topics were available to us. These versions had drawings from a South African artist, depicting sceneries which were more fitting for South Africa. Following focus group discussions with caregivers and healthcare workers prior to the production of the Vaccine SB, they suggested that the SB will be more acceptable if books were developed to depict The Gambian setting. Local artists were engaged to assist with drawings to help in the production of the book. All voice overs were done by local media personnel who could speak the local languages well. This improved the acceptability of the tool by the communities.
The discussion could be more concise and focus more on interpretation rather than re-reporting the findings. Why do you think the SB worked/did not work? Why do you think participants had a high level of trust in the messages?	We have revised our discussion extensively in line with the reviewer’s comment.

Please check all references have all correct information.	We have checked to ensure that all references have the correct information.
---	---

VERSION 2 – REVIEW

REVIEWER	Ilaria Montagni Bordeaux Population Health U1219 - University of Bordeaux and Inserm, France
REVIEW RETURNED	05-Nov-2020

GENERAL COMMENTS	Dear authors, thank you for having taken into account my comments. I have re-read the paper and found the corresponding modifications (attention: the pages and paragraphs you mention in the point-by-point table cannot be found in the paper). I am not totally convinced by your definition of vaccine literacy. I would not mention this term throughout the paper, but just talking about "vaccine knowledge". However, your new sentence in the discussion clarify this concept. There are still some minor typos that can be handled during the editing phase.
--

REVIEWER	Jo Durham Queensland University of Technology, School of Public Health & Social Work
REVIEW RETURNED	08-Dec-2020

GENERAL COMMENTS	Overall, the paper is improved. The dot point on strengths and limitations does not seem to relate to strengths/limitations. The speaking book is the object of the evaluation/study – I think strengths/limitations should relate to how the methods employed provided relevant insights into the questions (and limitations) I'm not convinced the study design reaches the definition of mixed methods – I would prefer to see the study called a study which used qual and quant methods Data collection methods – not clear what the "wide literature search" was used for With no narrative the themes from the open-ended questions do not seem to add anything paper – and their purpose is unclear Given the long list of barriers included in the introduction the discussion should acknowledge while useful and possibly a first step to behaviour change the intervention (i.e. the speaking book) does not address the multiple barriers families face in getting their children vaccinated as well as the health system barriers identified by the HCW such as lack of staff Paragraph p.19 line 19 seem underdeveloped – the point of the paragraph and how it adds to the general narrative is unclear I don't think the paper needs to be seen by a statistician but should add I am not a statistician
---

VERSION 2 – AUTHOR RESPONSE

Reviewer: 2	Authors response
Dear authors, thank you for having taken into account my comments. I have re-read the paper and found the corresponding modifications (attention: the pages and paragraphs you mention in the point-by-point table cannot be found in the paper). I am not totally convinced by your definition of vaccine literacy. I would not mention this term throughout the paper, but just talking about "vaccine knowledge". However, your new sentence in the discussion clarify this concept. There are still some minor typos that can be handled during the editing phase.	We have rephrased the point in the introduction section to read: "The caregivers' knowledge and overall vaccine knowledge influence their decision to access immunization services for their infants."

Reviewer: 3	Authors response
Overall, the paper is improved.	We acknowledge with thanks the reviewer's comment.
The dot point on strengths and limitations does not seem to relate to strengths/limitations. The speaking book is the object of the evaluation/study – I think strengths/limitations should relate to how the methods employed provided relevant insights into the questions (and limitations).	We have changed the dot points on strengths and limitations to reflect your suggestions in the Strengths and Limitation section. It is also reflected upon in the discussion section at the end of the paper.
I'm not convinced the study design reaches the definition of mixed methods – I would prefer to see the study called a study which used qual and quant methods.	We have reworded the title and have reflected this in the methods section which now reads as follows: "We conducted a study which used quantitative and qualitative methods to enrol caregivers and their infants attending immunization clinics in 15 purposively selected Primary Health Care facilities (PHCs) across four regions of The Gambia."
With no narrative the themes from the open-ended questions do not seem to add anything paper – and their purpose is unclear.	We believe that the themes from the open-ended questions further explore the experiences of caregivers using the speaking book and are of value to the field, especially for social scientists who might take an interest in our approach. Other reviewers and the editor did not question this approach and it is common to cite the themes and quotes in qualitative research. If the editor feels we should remove them, we can do so.

Given the long list of barriers included in the introduction the discussion should acknowledge while useful and possibly a first step to behaviour change the intervention (i.e. the speaking book) does not address the multiple barriers families face in getting their children vaccinated as well as the health system barriers identified by the HCW such as lack of staff.	We certainly acknowledge that the Speaking Book can only be one of the many tools needed to change behaviour and we have added a sentence in line with the reviewer's statement to the discussion. We are very aware that families face a whole range of barriers to the uptake of immunisations and health services in general - the Speaking Book isn't going to be able to overcome in isolation. We have additionally emphasised this in the discussion paragraph just before the conclusions.
Paragraph p.19 line 19 seem underdeveloped – the point of the paragraph and how it adds to the general narrative is unclear.	We agree and have removed this paragraph from the discussion as it does not add value.
I don't think the paper needs to be seen by a statistician but should add I am not a statistician.	We acknowledge with thanks the reviewer's comment.

VERSION 3 – REVIEW

REVIEWER	Jo Durham Queensland University of Technology, Australia
REVIEW RETURNED	21-Jan-2021
GENERAL COMMENTS	None